# TO ERR IS MACHINE: VULNERABILITY DETECTION CHALLENGES LLM REASONING

## ABSTRACT

In this paper, we present a challenging code reasoning task: vulnerability detection. Large Language Models (LLMs) have shown promising results in natural-language and math reasoning, but state-of-the-art (SOTA) models reported only 54.5% Balanced Accuracy in our vulnerability detection evaluation, even those models pre-trained on large amounts of source code. Our error analysis on LLM responses shows that the models struggle to reason about the code semantics relevant to identifying vulnerabilities, especially subtle semantic differences caused by small textual changes. We explored prominent models and training settings to understand their effects on vulnerability detection performance — including better prompts, larger models, more pre-training data, and fine-tuning — but none led to significant improvements. This raises the question of whether simply scaling training data and model size will allow us to "solve" complex code reasoning tasks like vulnerability detection, or if a fundamental shift in modeling and training techniques is required. We also explored adding domain knowledge to prompts; although it helped certain models understand some code semantics, vulnerability detection requires multi-step reasoning, and these models still failed in steps, such as reasoning about variable relations. Our results suggest that new models, new training methods, or more execution-specific pretraining data may be needed to conquer vulnerability detection. We speculate that auto-regressive pre-training on source code may not effectively extract code semantics, especially on the current pretraining mixtures, in which execution data is scarce. Success on vulnerability detection as a code reasoning task can benefit many areas of software engineering such as debugging, test input generation, and program repair. Our code and data are available at `https://figshare.com/s/78fe02e56e09ec49300b`.

## 1 INTRODUCTION

Thousands of new software vulnerabilities are discovered each year, costing users and companies millions of dollars (NIST, 2024a). This makes vulnerability detection critically important for software security. Since Devign in 2019 Zhou et al. (2019), many deep learning approaches have been proposed to predict the presence of vulnerabilities, but model performance has not breached 70% F1 score on realistic datasets Chen et al. (2023). In this paper, we show that though existing LLMs achieve impressive results on math, natural language, code reasoning and code generation tasks Talmor et al. (2019); Cobbe et al. (2021); Gu et al. (2024); Chen et al. (2021) they struggle to detect vulnerabilities (Section 2). We show that vulnerability detection is a complex code reasoning challenge, requiring both multi-step analysis and an accurate understanding of code semantics. This paper makes the case that vulnerability detection presents a compelling new target task for the ML community; solving it could significantly impact related software engineering tasks, such as debugging, test input generation, and program repair, thereby enhancing developer productivity. Furthermore, improving LLMs' ability to reason about code and identify vulnerabilities could potentially drive progress in broader reasoning tasks.

As shown in Figure 1, to detect a vulnerability, a developer first needs to accurately locate the statements relevant to a potential vulnerability. Second, a developer must understand the semantics of those relevant statements, which requires domain knowledge, such as recognizing bounds/NULL checks and understanding the effects of string, pointer, and arithmetic operations. Sometimes only a single operator distinguishes vulnerable and non-vulnerable versions of code, and effective vulnera-

```
1  get_next_file(FILE *VFile, char *ptr) {
2    char *ret;
3    ret = fgets(ptr, PATH_MAX, VFile);
4    if (!ret) return NULL;
5
6  -  if (ptr[strlen(ptr) - 1] == '\n')
7  -    ptr[strlen(ptr) - 1] = '\0';
8  +  size_t len = strlen (ptr);
9  +  if (len > 0 && ptr[len - 1] == '\n')
10 +    ptr[len - 1] = '\0';
11   return ret;
12 }
```

```
1  mrb_class_real(struct RClass* cl) {
2    if (cl == 0) return NULL;
3    cl->super = NULL;
4    // ...
5    while ((cl->tt == MRB_TT_SCLASS) || (cl->tt
↪    == MRB_TT_ICLASS)
6    ) {
7      cl = cl->super;
8  +    if (cl == 0) return NULL;
9    }
10   return cl;
11 }
```

(A) Buffer Overflow (BOF). To detect this vulnerability in the vulnerable version, the model/developer takes several reasoning steps: (step 1) identify the BOF-relevant statements, e.g., buffer allocation in line 3 and access in line 6; (step 2) understand that the allocated buffer may be empty depending on user input and that `strlen(ptr)` returns 0 in line 6 if the buffer is empty; (step 3) connect the facts, recognizing that if the buffer is empty, then line 6 will access index −1, causing a BOF. In the patched version, the model/developer should recognize in step (2) that line 9 checks the length of the buffer before accessing it, and therefore, step 3 concludes that this vulnerability is not exploitable.

(B) Null-Pointer Dereference (NPD). To detect this vulnerability in the vulnerable version, the model/developer takes several reasoning steps: (step 1) identify the relevant statements, e.g. the assignments `cl->super = NULL` in line 3 and `cl = cl->super` in line 7, and dereference of `cl` in line 5; (step 2) understand that in line 3, `cl->super` is set to NULL; (step 3) connect the facts, recognizing that after assigning `cl` to NULL in line 7, it will be dereferenced when the loop condition is evaluated in line 5, causing a NPD. In the patched version, the model/developer should recognize in step (2) that line 8 checks if `cl` is NULL and returns safely, thus there is no vulnerability.

FIGURE 1. Examples of vulnerability detection as a complex code reasoning task. Diffed lines (+/-) show the lines changed to patch the vulnerability.

bility detection requires understanding these nuances of program semantics. Finally, the developer must logically connect the individual facts about the statements to infer whether a vulnerability exists. This last step requires reasoning about the ranges of values and the temporal relations of symbolic variables, and then comparing them to the application's security policy, which is often implicit.

These steps are challenging for LLMs, both individually and in combination. We studied 14 SOTA LLMs and 7 prompting methods on SVEN (He & Vechev, 2023), a high-quality, real-world dataset consisting of 386 vulnerable functions and their corresponding fixed versions, covering 772 programs. We found that *all models and prompts performed close to random guessing (50-55% Balanced Accuracy)* (Section 2). Even GPT-4, a SOTA model, couldn't distinguish vulnerable code from its fixed version for 67.4% cases.

After manually analyzing 300 of the LLM responses (Section 3), we found errors occurring at all three steps of the reasoning process. For instance, in step 1, localization, the models frequently (50% of inspected functions) failed to recognize bounds or NULL checks, resulting in false positives. Explicit marking of bounds checks is easily done by humans but seems to be difficult for LLMs to recognize. In step 2, LLMs misinterpreted string, pointer, and integer operations in 10%, 6%, and 8% of functions, respectively. Understanding bounds/NULL checks and the operations requires a precise understanding of code execution semantics, which LLMs generally struggle with Gu et al. (2024); our results confirm this finding and further indicate which structures were most challenging. We attribute the models' lack of understanding of code semantics, even after using various prompting methods, to two key factors: (1) the models may have limited exposure to execution data during pre-training, which restricts their ability to learn semantics directly – although LLMs might acquire some semantic understanding indirectly from simple executions aligned with code text, or developer's discussions about semantics; and (2) the current autoregressive pre-training methods face inherent difficulty of learning execution semantics from code text alone. This is likely why we observe that scaling up model size or dataset volume, and performing fine-tuning, did not significantly improve performance (Sections 3.1 and 3.2); since the necessary data are not in the dataset, it is unlikely that LLMs can learn this complex reasoning via scaling alone. Annotating code semantics in prompts reduced some of these errors in certain models (Section 3.3), but determining vulnerabilities require

a multi-step analysis. There are errors in other reasoning steps can further prevent the detection. We show that in step 3, LLMs frequently failed at multi-step logical reasoning, leading to inconsistent or non-sequential inferences in 9% of responses.

To the best of our knowledge, this paper is the first to utilize vulnerability detection to systematically explore the capabilities of existing LLMs to reason about complex code properties. Ullah et al. (2023) compared GPT-4's responses with human-written vulnerability summaries using metrics like *BLEU*, *ROUGE*, and *GPT-4 evaluations*, but did not delve into the specific failure modes which occurred in responses. Yu et al. (2024); Nong et al. (2024) examined GPT-4 and GPT-3.5's responses about vulnerabilities but did not perform systematic studies on a set of models, and on the impact of model sizes, training data, training methods, and adding domain knowledge. We classified the errors based on the challenges of reasoning steps, resulting in categories which are more fine-grained and actionable; we explored mitigating a specific type of error using a prototype with CoT-Annotations, as discussed in Section 3.3.

In summary, we make the following contributions:

(1) We clarify vulnerability detection as a complex reasoning challenge;

(2) We demonstrate that current SOTA LLMs severely underperform in vulnerability detection, achieving only 50-55% balanced accuracy at best;

(3) Through manual analysis of hundreds of LLM responses, we reveal that LLMs struggle with all stages of reasoning, particularly in understanding semantics of statements involving bounds/NULL checks, string operations, and pointer handling, which contributes significantly to their poor performance;

(4) We show that these reasoning failures and low performance cannot be easily mitigated by increasing model size, improving training data, and applying fine-tuning, even when the model is provided with domain-specific knowledge.

The fact that vulnerability detection exposes the limitations in current models' abilities to reason about vulnerabilities, coupled with the availability of well-defined vulnerability data, makes vulnerability detection an ideal benchmark for evaluating and challenging LLM reasoning capabilities.

## 2   CAN LLMS EFFECTIVELY DETECT VULNERABILITIES?

**Prompts:** We used the baseline prompting methods including *Basic (zero-shot)* Fu et al. (2023) and *In-context (n-shot)* Liu et al. (2023b); Zhou et al. (2024) prompts. We used a system prompt to set the context and instructions including the vulnerability definition (MITRE, 2024) and the program source code (see Section A for details).

We designed three additional prompting methods that leverage the metadata available in vulnerability datasets to encourage reasoning and provide the domain knowledge to the model, namely: (1) *In-context examples from contrastive pairs (Contrastive)*, which uses pairs before and after a bug-fix as in-context examples, with the goal of instructing the model the fine-grained differences which caused the bug, (2) *Chain-of-Thought from CVE descriptions (CoT-CVE)*, which uses CVE bug reports (CVE, 2024) from the Big-Vul dataset (Fan et al., 2020), prompting the model to respond with the explanations of vulnerability, and (3) *Chain-of-Thought from static analysis (CoT-StaticAnalysis)*, which adapts vulnerability proofs output by a static analyzer (Calcagno & Distefano, 2011) as reasoning steps for the example response, conditioning the model to reason step-by-step. We obtained the proofs from the D2A dataset Zheng et al. (2021) (see Section A for details).

**Models:** We evaluated 14 LLMs which are the SOTA in code generation, based on Zhao et al. (2023); Liu et al. (2023a) and the HumanEval leaderboard (PapersWithCode, 2023) (as of March 2024). The models include LLAMA 2 (Touvron et al., 2023), Code LLAMA (Roziere et al., 2023), StarCoder (Li et al., 2023b), StarChat (Tunstall et al., 2023), StarCoder2 (Lozhkov et al., 2024), StarChat2 (HuggingFaceH4 Team, 2024), Mistral (Jiang et al., 2023), Mixtral (Jiang et al., 2024), MagiCoder (Wei et al., 2023), Wizardcoder (Luo et al., 2023), DeepSeek-Coder(Guo et al., 2024), GPT-3.5 (OpenAI, 2023), GPT-4 (OpenAI, 2024), and Gemini 1.0 Pro (Gemini Team, 2023). See Section B for details.

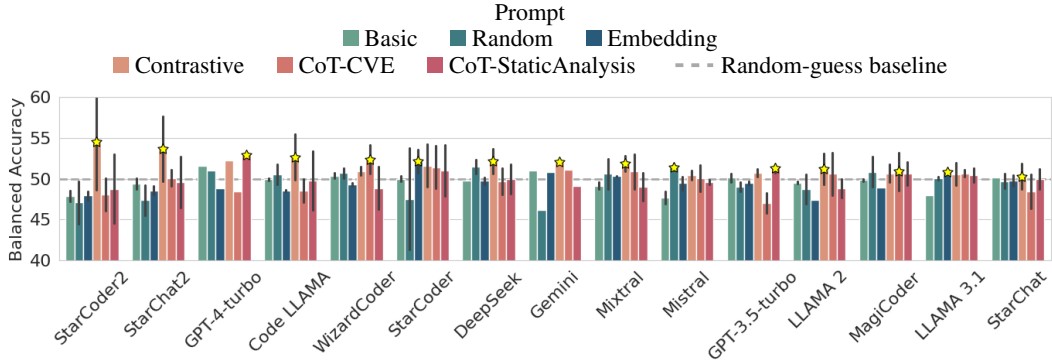

FIGURE 2. Vulnerability detection performance. Bar height shows the average performance of three random seeds and error bars show standard deviations; stars (⭐) mark the best-performing prompt for each model.

| Model (params) | SVEN Vuln. detection | HumanEval Code gen. | CruxEval Code execution | GSM8k Math | CSQA NL reasoning |
|---|---|---|---|---|---|
| StarChat | 50.9 | - | - | - | - |
| LLAMA 2 (34b) | 51.2 | 22.6 | - | 42.2 | - |
| StarCoder (15.5b) | 52.2 | 45.8 | 34.2 | - | - |
| Mistral (7b) | 51.4 | 30.5 | 34.3 | 36.5 | 62.5 |
| Mixtral (8x7b) | 51.9 | 40.2 | 40.5 | 58.4 | 86.7 |
| Code LLAMA (34b) | 52.6 | 48.8 | 42.4 | - | - |
| WizardCoder (33b) | 52.4 | 59.8 | 43.4 | - | - |
| MagiCoder (7b) | 50.9 | 70.7 | 44.4 | - | - |
| StarCoder2 (16b) | 54.5 | 46.3 | 47.1 | - | - |
| GPT-3.5-turbo | 51.8 | 64.9 | 49.4 | 57.1 | 85.5 |
| DeepSeek (33b) | 52.1 | 69.2 | 49.9 | - | - |
| GPT-4-turbo | 52.9 | 87.1 | 68.7 | 87.1 | 95.3 |
| Gemini 1.0 Pro | 52.1 | 67.7 | - | 86.5 | 84.7 |
| StarChat2 | 53.6 | 71.3 | - | - | - |

TABLE 1. Performance on vulnerability detection vs. NL/math reasoning, code generation, and code execution. Sources for code, math, and NL reasoning performance are cited in Section C.

**Benchmark and Metrics:** We used the SVEN dataset (He & Vechev, 2023), which contains 772 vulnerable and fixed functions from real-world C/C++ projects (average length = 168 lines). Existing vulnerability datasets like PrimeVul (Ding et al., 2025) are useful for fine-tuning but suffer from label noise, reducing the reliability of evaluations; in contrast, SVEN aggregates and manually vets vulnerabilities from multiple benchmarks, with 94% reported label accuracy (Ding et al., 2025). Because the commonly used F1 score can bias towards models which predict vulnerable more often (Zhou et al., 2024), we used *Balanced Accuracy* (Brodersen et al., 2010) (defined as $(\frac{correct_{vul}}{examples_{vul}} + \frac{correct_{nvul}}{examples_{nvul}})/2$) to evaluate the models.

**Results:** Figure 2 shows the performance of the baseline methods and our proposed prompts. While our new prompts slightly improved the best-case performance for 11 out of 14 models, with Contrastive prompts enhancing 8 out of 14, none of the models or prompts exceeded the random-guessing baseline (Balanced Accuracy = 50) by more than 5% Balanced Accuracy. In doubt of whether the complexity of the real-world code is the main challenging factor, we studied simple code examples (25 lines per function on average) from the CWE and SARD databases (MITRE, 2024; NIST, 2024b) and found that the models still did not predict simple functions correctly, reporting 42-67% Balanced Accuracy across all the models (see Section D). Table 1 compares the models' vulnerability detection performance with their performance in other domains. While new models have made steady advances in code generation Chen et al. (2021), code execution Gu et al. (2024), NL reasoning Talmor et al. (2019), and math reasoning Cobbe et al. (2021), their vulnerability detection performance has not increased in step, remaining close to the random-guess baseline. This result

implied that the until-now successful strategies of scaling model size and training data have not yet proven to be sufficient to solve vulnerability detection; to further confirm this, we investigated further in Sections 3.1 and 3.2.

TABLE 2. Models' abilities to distinguish pairs of vulnerable and non-vulnerable examples. Cell values display the number and percentage of pairs in each category.

| | | Distinguished | |
| Model | Can't Distinguish | Both Correct | Both Wrong |
| --- | --- | --- | --- |
| StarChat | 86.1% | 7.9% | 6.1% |
| DeepSeek | 82.5% | 6.3% | 11.2% |
| StarCoder | 82.1% | 12.5% | 5.4% |
| GPT-3.5-turbo | 80.9% | 11.3% | 7.8% |
| LLAMA 2 | 76.5% | 15.6% | 8.0% |
| MagiCoder | 75.2% | 11.9% | 12.9% |
| Mixtral | 67.8% | 18.3% | 13.9% |
| GPT-4-turbo | 67.4% | 18.9% | 13.7% |
| Gemini | 64.4% | 19.1% | 16.5% |
| Mistral | 61.8% | 20.6% | 17.6% |
| StarChat2 | 61.4% | 21.0% | 17.6% |
| StarCoder2 | 57.5% | 19.0% | 23.5% |
| Code LLAMA | 57.3% | 22.3% | 20.4% |
| WizardCoder | 55.0% | 23.8% | 21.1% |
| Average | 69.7% | 16.3% | 14.0% |

Table 2 presents our results on the models' capabilities of distinguishing pairs of vulnerable code and its fixed version. In the table, under Column *Can't Distinguish*, we show that, on average across all the models, 69.7% of pairs could not be distinguished, indicating that the models do not understand the nuanced semantics of the vulnerability. Some models/prompts were better than average, but at best, 55.0% of pairs could not be distinguished. The *Both Correct* and *Both Wrong* columns indicate that the models predicted both versions correctly in some instances (16.3% of pairs), but there were also cases (14.0% of pairs) where the models predicted both versions incorrectly.

# 3   WHY DO LLMS FAIL TO REASON ABOUT VULNERABILITIES?

We manually inspected 300 vulnerable predictions (covering 100 programs) from 14 models, regarding the vulnerability reasoning steps, including locating and understanding the semantics of statements related to vulnerability decisions, as well as the logical reasoning that can integrate variable values and relations, and compare them with the security policy. To reduce subjectivity, our manual inspection used independent ratings from three authors, following Islam et al. (2019), and adopted best practices for inductive coding (Saldaña, 2021). See Section F.3 for the details of our protocol.

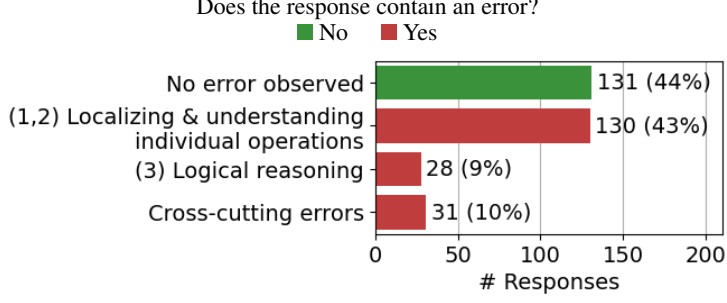

FIGURE 3. Error categories observed in responses from all LLMs. Bar width shows the number of responses that contained the category of error. One response can contain more than one type of error.

Figure 3 summarizes the errors. The results show that LLMs had some successes in reasoning, with 44% of responses containing no observed errors; however, still more than half of the responses

TABLE 3. Error analysis from 300 responses covering 100 programs. We analyzed the errors manually using the rubric and inter-rater agreement procedure detailed in Section F.

| Reasoning step | Error | Count |
|---|---|---|
| (1,2) Localizing and understanding statements related to vulnerability | Misunderstood Bounds/NULL check | 80/159 (50%) |
| | Misunderstood string operation | 3/29 (10%) |
| | Misunderstood arithmetic operation | 8/96 (8%) |
| | Misunderstood pointer operation | 9/147 (6%) |
| | Misunderstood alloc/free operation | 4/81 (5%) |
| | Misunderstood index operation | 1/60 (2%) |
| | Misunderstood execution order | 11 |
| | Improper assumption | 8 |
| | Misunderstood syntax | 6 |
| | Total | 125 |
| (3) Logical reasoning | Faulty implication ($\Rightarrow$) | 14 |
| | Inconsistent ($\perp$) | 14 |
| | Total | 28 |
| Cross-cutting errors | Hallucination | 15 |
| | Memorization | 11 |
| | Repetition | 5 |
| | Total | 31 |

contained an error in at least one step. LLMs made errors on localizing and understanding individual statements for 43% examples; this causes them to make faulty inferences about the effects of the code and flag potential vulnerabilities in safe code. LLMs also made cross-cutting errors such as hallucination and repetition 10% of the time and made incorrect logical inferences 9% of the time.

The LLMs frequently made errors related to several specific code structures, shown in Table 3. For example, out of 159 responses explaining bounds/NULL checks, 80 (50%) were incorrect. The semantics of bounds/NULL checks are critically important for determining whether several pertinent vulnerabilities exist, including buffer overflow, null-pointer dereference, and use after free. Such checks often follow predictable code patterns and thus are relatively simple for developers and static analysis tools to identify — we used static analysis to recognize them in Section 3.3 — however, the LLMs often failed to recognize these patterns. In addition, the models face challenges of understanding the semantics of operations; for example, the models incorrectly interpreted 10% string operations and 8% of arithmetic operations, which are necessary for reasoning about buffer overflow and integer overflow vulnerabilities.

FIGURE 4. Missed Bounds/NULL check.

In Figure 4, StarChat reported that there is an unchecked null-pointer dereference at line 5 (`p->lineinfo[oldpc]`). However, it overlooked the safety check at line 2, where null values for `p->lineinfo` are handled safely. Figure 5 provides an example of a *Misunderstood arithmetic operation* error. GPT-4 correctly identified the bounds-check at line 6, which had been added by developers to prevent overflows (MITRE, 2024). However, the LLM failed to reason about the

FIGURE 5. Misunderstood arithmetic operation.

**Source code** (label: non-vulnerable, prediction: vulnerable)

```
1  void AllocateDataSet(cmsIT8* it8) {
2    TABLE* t = GetTable(it8);
3    if (t -> Data) return;  // Already allocated
4    t-> nSamples = atoi(cmsIT8GetProperty(it8, "NUMBER_OF_FIELDS"));
5    t-> nPatches = atoi(cmsIT8GetProperty(it8, "NUMBER_OF_SETS"));
6    if (t -> nSamples < 0 || t->nSamples > 0x7ffe || t->nPatches < 0
   ↪ || t->nPatches > 0x7ffe)
7      SynError(it8, "AllocateDataSet: too much data");
8    else {
9      t->Data = (char**)AllocChunk(it8, ((cmsUInt32Number)t->nSamples
   ↪ + 1) * ((cmsUInt32Number)t->nPatches + 1) * sizeof(char*));
10     if (t->Data == NULL) {
11       SynError(it8, "AllocateDataSet: Unable to allocate data
   ↪ array");
12     }
13   }
14 }
```

**Model response** (GPT-4)

Both `t->nSamples` and `t->nPatches` are user-controlled and can be large. When these values are incremented by 1 and then multiplied by each other and further multiplied by `sizeof(char*)`, the result can overflow a 32-bit integer if the values are sufficiently large...
Before performing the allocation, the function checks if `nSamples` and `nSamples` are within a certain range (`0` to `0x7ffe`). However, this check does not prevent overflow. For example, values close to the upper limit can still cause the multiplication result to wrap around, leading to a smaller allocation than required.

calculation of the argument value to `AllocChunk` at line 9. Given the upper bound of `0x7fff` for `nSamples+1` and `nPatches+1`, even the maximum values would not cause an overflow in an unsigned integer ($0x7fff * 0x7fff * 8 = 0xfff80008$), so the LLM's alert is a false positive.

As a follow-up to the error analysis, we conducted a form of *natural experiment* Wikipedia (2024) to compare various LLMs and assess whether prominent training strategies improved vulnerability performance. This study design enabled us to evaluate each training strategy independently while controlling other variables. We compared models of different *sizes* (Section 3.1) and models trained with varying data and training methods, including *increased training data volume*, *code vs. NL training data*, *instruction fine-tuning*, and *adapter fine-tuning* (Section 3.2). We also investigated the use of external tools (Section 3.3) to add domain knowledge targeting the types of reasoning errors we found in Table 3.

## 3.1 DOES MODEL SIZE MATTER?

We evaluated several models which released different sizes: LLAMA 2 (7b, 13b), Code LLAMA (7b, 13b, 34b), Mistral 7b vs. Mixtral 8x7b, and DeepSeek-Coder (1.3b, 6.7b, and 33b). Figure 6 shows that model performance did not significaly improve by scaling up the model size, and we found that there was no statistical correlation between model size and performance ($R^2 = 0.02$, $p = 0.72$). We manually analyzed the responses using the methodology in Section 3 and found that all models had error rates similar to those shown in Figure 3, although larger models were better at following in-context prompts. For example, Code LLAMA 7b, often analyzed the in-context examples instead of the queried example; this error happened less frequently with Code LLAMA 13b, and not at all with Code LLAMA 34b. This aligns with previous results (Wei et al., 2022a) showing that in-context learning is an emergent property of larger models.

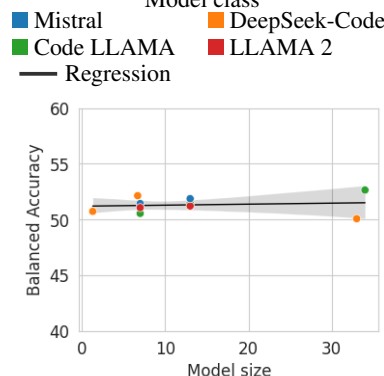

FIGURE 6. Larger models did not improve on vulnerability detection.

## 3.2 DO MODEL TRAINING DATA & METHODS MATTER?

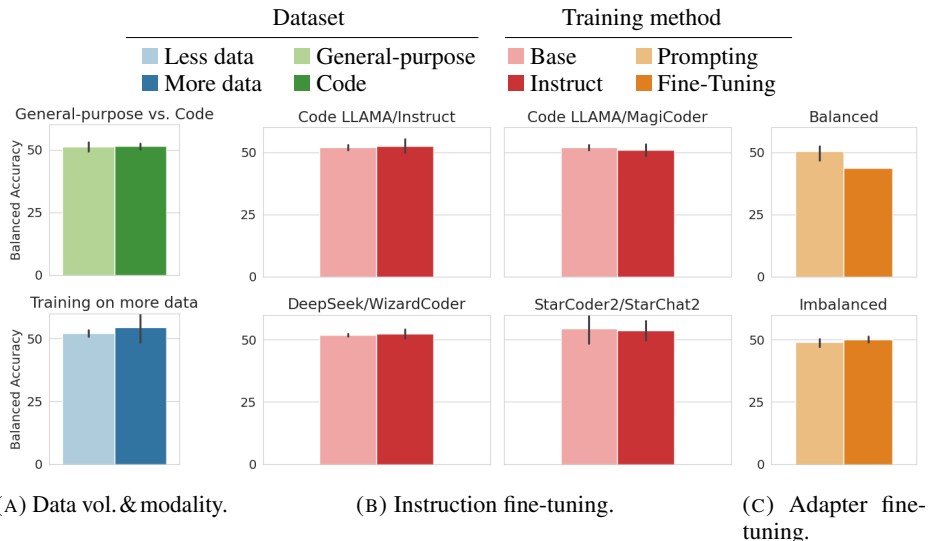

FIGURE 7. Expanding the training dataset and incorporating fine-tuning had minimal impact on vulnerability detection capability.

■ **Figure 7a (top), General-purpose vs. code training data:** Models trained mainly on natural language may lack the knowledge of code seen in models which have been fine-tuned on code. This raises the question: do models specialized for code outperform general-purpose models? To explore this, we compared LLAMA 2, designed as a general-purpose chat assistant (Touvron et al., 2023), against Code LLAMA, which was initialized from the base weights of LLAMA 2 and further fine-tuned on code (Roziere et al., 2023). Figure 7a (top) shows that code-specialized training did not substantially improve Code LLAMA's vulnerability detection capability.

■ **Figure 7a (bottom), Training on more data:** HuggingFace's Bigcode team released StarCoder and its updated version, StarCoder2, with the primary difference being that StarCoder2 trained on more than twice as much code (Lozhkov et al., 2024). This setup provides a relatively controlled environment to assess the impact of this additional training data. Figure 7a (bottom) indicates that scaling up the dataset resulted slightly improved StarCoder2's vulnerability detection capability, but yielded only a 4% improvement.

■ **Figure 7b, Instruction fine-tuning :** Instruction fine-tuning can improve the truthfulness and relevance of responses (Ouyang et al., 2022), as well as performance and generalization (Chung et al., 2024). This leads us to ask: do instructed models perform better than their base counterparts? We compared the base versions of DeepSeek Coder, StarCoder2, and Code LLAMA against their instruction fine-tuned counterparts, namely WizardCoder, StarChat2, and Code LLAMA-Instruct/MagiCoder respectively, and found no substantial difference in performance (Figure 7b), indicating that instruction fine-tuning did not improve vulnerability detection performance, even though our vulnerability detection prompts are tailored for instruction-tuned models.

■ **Figure 7c, Adapter fine-tuning:** We fine-tuned the StarCoder2 7b model using the vulnerability dataset PrimeVul Ding et al. (2025) (the cleanest supervised dataset large enough for fine-tuning) and compared it with the base 7b model using the Basic prompt on SVEN. We evaluated the models on balanced and imbalanced label distributions, following Chakraborty et al. (2022). Figure 7c shows that finetuning did not substantially improve the model's performance in either case; the fine-tuned model performed worse on balanced data and only 1.2% better on imbalanced data.

## 3.3 DOES ADDITIONAL DOMAIN KNOWLEDGE HELP?

Table 3 indicates that one of the important challenges that prevented LLMs from detecting vulnerabilities is their incapability of understanding *bounds/NULL checks* and *pointer operations*. Thus, we

---

**CoT-Annotations prompt for the source code in Figure 1b**

Code Explanation: The pointer `cl` is passed as parameters in this function, which could be null. In line `cl->super = NULL`, the pointer `cl->super` is assigned to NULL. In line `while ((cl->tt == MRB_TT_SCLASS)...`, the pointer `cl` is dereferenced. The function has a NULL check in line `if (cl == 0) return NULL;` for the pointer `cl`.

Instructions: For the Query Example only, think step-by-step using these steps, then give an answer based on the Code Explanation. Assume pointers which are passed in will not cause a null-pointer dereference.
1. identify which locations pointers are likely to be null.
2. identify which of these are dereferenced.
3. identify which of these dereferences of potentially-null pointers are checked and handled. Filter these out.
4. if any unchecked dereferences remain, then the function is vulnerable. Otherwise, the function is not vulnerable.

---

**Example response**

Reasoning: The function dereferences the pointer `cl` without checking if it is NULL. `cl` may be assigned to NULL in the line `cl = cl->super`. If `cl` is NULL, the function will crash when it tries to access the field `cl->tt` in the loop condition.
Answer: Yes, the function is buggy.

---

FIGURE 8. Example CoT-Annotations using the NPD vulnerability from Figure 1b. CoT-Annotations uses static analysis to annotate null assignments, parameters, dereferences, and null-checks (top) and provides reasoning example responses to encourage reasoning (bottom).

developed *Chain-of-Thought with Annotations (CoT-Annotations)*, shown in Figure 8. We introduced an external static analysis tool which annotates the code to highlight possible NULL assignments to pointers, NULL checks, on pointers, and dereferences of pointers. These annotations provide the exact information that defines the vulnerability and that a domain expert would use to identify NULL-pointer dereference vulnerabilities. We integrated such knowledge into the prompt and evaluated performance on detecting Null-Pointer Dereference (NPD) vulnerabilities for the models we studied above, as a case study. As a quality measure, we manually verified the static analysis output and excluded incorrect annotations caused by heuristic errors.

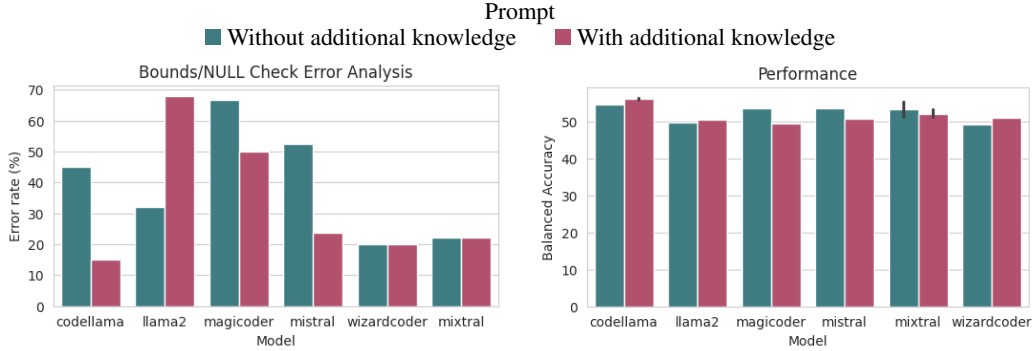

(A) Reasoning errors related to bounds/NULL checks.

(B) Vulnerability detection performance.

FIGURE 9. Domain knowledge is helpful for one step but not much for overall performance. Some models, e.g., CodeLLAMA, respond to domain knowledge better: case study on NPD vulnerabilities.

By analyzing a sample of 198 responses[1] with/without annotations, we observed that CoT-Annotations reduced the errors of bounds/NULL checks recognition by 15-70% for Code LLAMA, MagiCoder,

---

[1]This sample represents the intersection of vulnerable responses across all six models, with a maximum of 25 responses per model. We used the error categories established in Section 3, with each rater analyzing one-third of the responses (each response was reviewed by a single rater due to time constraints).

and Mistral, shown in Figure 9a; however, these models still missed 23-67% of bounds checks. We also observed that the improvement of understanding bounds/NULL checks did not significantly improve the models' performance (see Figure 9b). We speculate that this is because there are other blocking issues such as logical reasoning about relations of variables.

## 4 RELATED WORK

Recent studies have initiated investigation into the usage of LLMs for vulnerability detection, using zero-shot prompting Purba et al. (2023); Fu et al. (2023), in-context learning Gao et al. (2023); Liu et al. (2023b); Chan et al. (2023), and fine-tuning Shestov et al. (2024); Yusuf & Jiang (2024); Yang et al. (2024). Several papers have utilized chain-of-thoughts (CoT), such as "Let's think step-by-step" Li et al. (2023a); Feng & Chen (2024); Sun et al. (2024), multi-step prompts Ullah et al. (2023); Yu et al. (2024), and generic information such as CFG, DFG, PDG, and API calls Zhang et al. (2023); Nong et al. (2024); Khare et al. (2023); Ullah et al. (2023). In this work, we studied the most common prompting methods and proposed four novel prompt approaches tailored for vulnerability detection, integrating information from bug-fix commits (contrastive pairs), CVE descriptions (CoT-CVE), static analysis reports (CoT-StaticAnalysis), and domain knowledge annotations (CoT-Annotations). We further studied the LLMs' capabilities to distinguish buggy and patched versions of code and studied the reasoning errors in their responses.

Several recent papers have analyzed errors in LLM-generated vulnerability detection responses. Ullah et al. (2023) used BLEU, ROUGE, and GPT-4 to automatically compare GPT-4's reasoning summaries with human-generated ones. Yu et al. (2024); Nong et al. (2024) examined 82-100 responses from GPT-4 and GPT-3.5, supporting our findings that the models struggled with correctness, logic and consistency in general. However, existing studies do not match the depth and breadth of ours. Our error classifications provide more actionable and detailed categories, enabling us to identify specific code structures and LLM weaknesses (see Table 3). Additionally, we analyzed factors such as model size, training data, and training strategies, providing cause for concern about future improvements from model scaling. To our knowledge, our study is the most comprehensive manual analysis of LLMs for vulnerability detection, including 14 models and manually analyzing 300 LLM responses with a rigorous multi-rater agreement protocol.

## 5 CONCLUSION

In this paper, we have show that vulnerability detection is complex, multistage reasoning task that current LLMs struggle to solve. We conducted a thorough study to show that the SOTA models and prompts performed only slightly better than random guessing. None of the model advancements we explored led to significant improvements, including increasing model size, expanding training data, and instruction/adapter fine-tuning. The models particularly struggled to distinguish between vulnerable and fixed versions of code, where small textual differences cause large changes in semantics. We demonstrated that external tools and domain knowledge helped somewhat with single-step reasoning, but did not significantly improve the models' performance, which depends on accurate multi-step reasoning. Our findings bring concerns about further research in this area, raising the question of whether auto-regressively pre-trained LLMs are a good fit for tasks which require deep understanding of code semantics. We suggest that a fundamental shift in modeling and training methods may be necessary in order to overcome the reasoning failures of current LLMs. We believe that solving code reasoning in vulnerability detection could help address many other challenging tasks in software engineering, such as debugging, code execution prediction, test input generation, and program repair. Reasoning-based models fine-tuned for inherent chain-of-thought, such as OpenAI o1 (OpenAI, 2024) and DeepSeek R1 (DeepSeek, 2024), offer a promising approach to this challenge. Furthermore, frequent localization/understanding errors, such as missing bounds/NULL checks in 50% of cases, demonstrate the need for additional context or scaffolding, as demonstrated by our CoT-Annotations prototype Section 3.3. We hope that our paper laid out some key insights and motivation for the machine learning community to solve this important challenge.

REPRODUCIBILITY STATEMENT

Our code and data, including the materials and tool used for error analysis, are available at this link: `https://figshare.com/s/78fe02e56e09ec49300b`. To encourage replication and transparency, we include several appendices detailing our prompts (Section A), the model ID's we used (Section B), the NL/math/coding benchmarks we cited (Section C), and our manual error analysis methodology (Section F).

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

SUPPLEMENTARY MATERIAL

## APPENDIX A   VULNERABILITY DETECTION PROMPTS

We explored several prompt designs, guided by model performance on a small dev set or the entire SVEN dataset.

**Basic (zero-shot) prompting (Fu et al., 2023):** We first designed a system prompt to set the context: "I want you to act as a vulnerability detection system", along with natural-language instructions: (1) *Basic query*: "Is the following function buggy? Please answer Yes or No." (We also tried "Is the following function vulnerable?"; however, our pilot study shows that it did not perform as well.) (2) *CWE list*: This prompt starts with "Does the following function contain one of the following bug types?", followed by a fixed list of bug types, e.g., "CWE-190: Integer Overflow"; (3) *Q/A*: Begin the query with "Question:" and begin the model's response with "Answer:". This conditions the model to respond in a question-answering mode.

**In-context ($n$-shot) prompting(Liu et al., 2023b; Zhou et al., 2024):** In this prompt, we provide examples of inputs and responses for in-context learning (Brown et al., 2020). The in-context examples condition the model to reply in the same format as the example responses (Xie & Min, 2022). The selection of in-context examples can impact the performance. We studied three settings: (1) randomly selecting examples, (2) using retrieval-augmented generation (RAG, Lewis et al. (2020)) to retrieve the examples that had similar CodeBERT (Feng et al., 2020) embeddings to the query example, and (3) selecting examples from *contrastive pairs* (see below for details). We explored several options for formatting in-context examples, such as appending all the examples in one chat-assistant message versus using separate messages (one message performed best) and varying the number of examples from 1 to 10 (6 performed best).

**In-context prompting based on contrastive pairs:** We formed *contrasting pairs* of in-context examples by providing the vulnerable version of the code (before the bug-fixing commit) and the fixed version (after the commit) as in-context examples in the same prompt. Since these two versions of the source code differ primarily in the portion related to the bug-fix, our intention is that this prompt template would highlight the cause of the bug and instruct the model to learn that the small differences in code can lead to different labels.

**In-context prompting based on CoT from CVE descriptions:** We designed "chain-of-thought" prompts by providing intermediate reasoning steps which lead to the answer, inspired by Wei et al. (2022b). We use in-context examples from the Big-Vul dataset (Fan et al., 2020), which includes the CVE bug reports. For vulnerable examples, we used the default in-context query and provide the chain-of-thought response. To produce such response, we adapt the descriptions in these bug reports to describe how the bug manifests. For example, CVE-2017-9211 Corporation (2024) describes the vulnerability, including the symptoms, attack surface, and variable involved:

```
The crypto_skcipher_init_tfm function in crypto/skcipher.c in the Linux
kernel through 4.11.2 relies on a setkey function that lacks a key-size
check, which allows local users to cause a denial of service (NULL
pointer dereference) via a crafted application.
```

We use this description as the CoT response and append "Therefore, the example is buggy" to complete the example response. For non-vulnerable examples, we provide the default in-context example query/response.

To ensure high-quality examples in spite of label noise Croft et al. (2023), we removed duplicate examples, excluded examples with overly long or short source code (50-750 tokens), and retained only those examples tied to vulnerabilities (i.e., bug reports) of the same types in SVEN.

**In-context prompting based on CoT from static analysis:** We also used the output buggy paths reported by the Infer (Calcagno & Distefano, 2011) static analysis tool to prepare the chains of thought prompt. Infer reports a single bug-triggering path for each example. The buggy path consists of a list of statements that can lead to the bug. We use in-context examples from the D2A dataset Zheng et al. (2021), which lists buggy paths from the Infer static analyzer Facebook (2024) for several open-source C++ projects. We convert the buggy paths to natural language descriptions and use them as the response. This is an example CoT response for a buffer overflow vulnerability:

```
1.  A buffer buf of size 10 is allocated at line 1.
2.  An index i is initialized to a value in the range [0, 100] at line 2.
3.  The index i is used to access buf at line 3.  This may exceed the
bounds of buf.
```

We append "Therefore, the example is buggy" to complete the example response. For non-vulnerable examples, we provide the default response.

To ensure high-quality examples in spite of label noise Croft et al. (2023), we removed duplicate examples, excluded examples with overly long or short source code (50-750 tokens), and selected examples with complete vulnerability proofs within the vulnerable function and removed those with incomplete reports.

The key difference between our CoT-Annotation prompt Section 3.3 and the CoT-StaticAnalysis prompt is that the former uses *lightweight, custom-built static analysis* to provide targeted information about specific vulnerability semantics, such as bounds and NULL checks—areas where LLMs struggle, as shown in Section 3. The latter relies on a *heavyweight, off-the-shelf commercial static analyzer* (Infer) to supply proofs for the vulnerabilities it is designed to handle, but cannot provide customized information.

## APPENDIX B  MODELS

We used the model sizes shown in Table 4 and the text generation parameters shown in Table 5 for our experiments. The model IDs are documented in our data package.

TABLE 4. 14 models we studied.

| Model | Parameters | Context Length |
|---|---|---|
| GPT-4 OpenAI (2024) | - | 128k |
| Gemini 1.0 Pro Gemini Team (2023) | - | 32k |
| GPT-3.5 OpenAI (2023) | - | 4k |
| Mixtral-MoE Jiang et al. (2024) | 45B | 8k∼128k |
| Code LLAMA Roziere et al. (2023) | 7B, 13B, 34B | 16k∼100k |
| LLAMA 2 Touvron et al. (2023) | 7B, 13B | 4k |
| WizardCoder Luo et al. (2023) | 33B | 2k |
| DeepSeek-Coder Touvron et al. (2023) | 1.3B, 6.7B, 33B | 4k |
| StarChat2 HuggingFaceH4 Team (2024) | 15.5B | 16k |
| StarCoder2 HuggingFaceH4 Team (2024) | 15.5B | 16k |
| StarChat Tunstall et al. (2023) | 15.5B | 8k |
| StarCoder Li et al. (2023b) | 15.5B | 8k |
| MagiCoder Wei et al. (2023) | 7B | 16k∼100k |
| Mistral Jiang et al. (2023) | 7B | 8k∼128k |

## APPENDIX C  BENCHMARKS FOR OTHER DOMAINS

We gathered the benchmark performance results for Table 1 from public benchmarks and from the papers associated with each model:

- CruXeval: `https://crux-eval.github.io/leaderboard.html`

- HumanEval: `https://paperswithcode.com/sota/code-generation-on-humaneval`

- GSM8k: reported in the models' papers (Touvron et al., 2023; Jiang et al., 2023; 2024; OpenAI, 2023; 2024; Gemini Team, 2023).

- CSQA: reported in the models' papers (Jiang et al., 2023; 2024; OpenAI, 2023; 2024; Gemini Team, 2023).

TABLE 5. Text generation parameters we used.

| Parameter | HuggingFace | OpenAI | Google |
|---|---|---|---|
| Top-$p$ | 0.9 | 1.0 | 1.0 |
| Temperature | 0.1 | 0.1 | 0.1 |
| Max. tokens generated | 512 | 512 | 512 |

## APPENDIX D   SIMPLE CWE EXAMPLES

Figure 10 shows a simple integer overflow vulnerability example. The CodeLLAMA model could detect this vulnerability. The response of the CodeLLAMA model is shown in Figure 11. The performances of all models on these simple examples are shown in Table 6.

**Integer Overflow**

```
1   int determineFirstQuarterRevenue()
2   {
3
4       // Variable for sales revenue for the quarter
5       float quarterRevenue = 0.0f;
6
7       short JanSold = getMonthlySales(JAN); /* Get sales in January */
8       short FebSold = getMonthlySales(FEB); /* Get sales in February */
9       short MarSold = getMonthlySales(MAR); /* Get sales in March */
10
11      // Calculate quarterly total
12      short quarterSold = JanSold + FebSold + MarSold;
13
14      // Calculate the total revenue for the quarter
15      quarterRevenue = calculateRevenueForQuarter(quarterSold);
16
17      saveFirstQuarterRevenue(quarterRevenue);
18
19      return 0;
20  }
```

FIGURE 10. A simple integer overflow example collected from CWE database. The summation of three short int numbers can be more than the value of the maximum short int number, which will introduce integer overflow vulnerability.

**Example response**

No, the function does not contain any of the following vulnerability types:
* CWE-190: Integer Overflow
* CWE-476: Null Pointer Dereference
* CWE-125: Out-of-bound Read
* CWE-787: Out-of-bound Write
* CWE-416: Use After Free

FIGURE 11. Code LLAMA's response to the simple example in Figure 10

TABLE 6. The performance of the studied models on simple CWE examples.

| Model | Performance |
|---|---|
| GPT-4 OpenAI (2024) | 65.78 |
| Gemini 1.0 Pro Gemini Team (2023) | 50.87 |
| GPT-3.5 OpenAI (2023) | 56.14 |
| Mixtral-MoE Jiang et al. (2024) | 61.40 |
| Code LLAMA Roziere et al. (2023) | 61.40 |
| LLAMA 2 Touvron et al. (2023) | 46.49 |
| WizardCoder Luo et al. (2023) | 51.75 |
| DeepSeek-Coder Touvron et al. (2023) | 66.67 |
| StarChat2 HuggingFaceH4 Team (2024) | 55.26 |
| StarCoder2 HuggingFaceH4 Team (2024) | 50.87 |
| StarChat Tunstall et al. (2023) | 50.00 |
| StarCoder Li et al. (2023b) | 41.52 |
| MagiCoder Wei et al. (2023) | 62.28 |
| Mistral Jiang et al. (2023) | 57.01 |

## APPENDIX E    PERFORMANCE BREAKDOWN BY BUG TYPE

Table 7 provides a breakdown of model performance by bug type.

TABLE 7. Performance breakdown by bug type.

| Model | CWE-125 (OOB Read) | CWE-190 (Integer Overflow) | CWE-416 (UAF) | CWE-476 (NPD) | CWE-787 (OOB Write) |
|---|---|---|---|---|---|
| Code LLAMA | 53.67 | 55.29 | 51.88 | 52.42 | 55.86 |
| Gemini | 55.08 | 54.65 | 57.73 | 55.51 | 52.68 |
| GPT-3.5-turbo | 53.83 | 53.16 | 51.59 | 51.09 | 54.5 |
| GPT-4-turbo | 54.53 | 54.65 | 53.17 | 55.61 | 53.57 |
| LLAMA 2 | 53.66 | 52.17 | 50.13 | 51.72 | 52.83 |
| MagiCoder | 53.67 | 52.04 | 55.91 | 52.44 | 57.23 |
| Mistral | 52.95 | 51.25 | 52.49 | 51.32 | 54.46 |
| Mixtral | 52.72 | 55.54 | 50.55 | 53.08 | 55.96 |
| StarChat | 50.84 | 52.06 | 52.58 | 52.52 | 50.92 |
| StarChat2 | 53.54 | 52.66 | 54.36 | 52.56 | 56.58 |
| StarCoder | 56.70 | 58.44 | 54.91 | 53.17 | 58.66 |
| StarCoder2 | 58.65 | 52.03 | 52.11 | 51.69 | 54.99 |
| WizardCoder | 53.01 | 53.85 | 55.38 | 53.55 | 57.55 |

## APPENDIX F   ERROR ANALYSIS METHODOLOGY

To support measurable and targeted improvements in model reasoning, we open-sourced our error analysis tool Anonymous (2024). Researchers can use this tool to quickly analyze a sample of LLM responses to measure whether and how much the LLMs improved on a specific reasoning error, as we demonstrated in Section 3.3. By offering concrete metrics and a practical tool to measure them, we aim to accelerate both advancements in LLM reasoning research and the adoption of reasoning-based models for vulnerability detection tasks.

### F.1   INTER-RATER AGREEMENT

We first analyzed 50 examples to create detailed error categories for each reasoning step. All three raters independently identified errors in the LLM responses, refining the protocol after processing ⅓, ½, and all of the data. We added new error categories when needed, and merged similar categories after analysis concluded. We measured inter-rater agreement using Fleiss' kappa ($\kappa$) Fleiss (1971), achieving 0.78 with 86% agreement. We resolved disagreements by majority vote, followed by discussion for the final categorization. After the categories were set in Section 3, we used one rater to analyze the responses reported in Sections 3.1 to 3.3.

### F.2   ERROR ANALYSIS UI

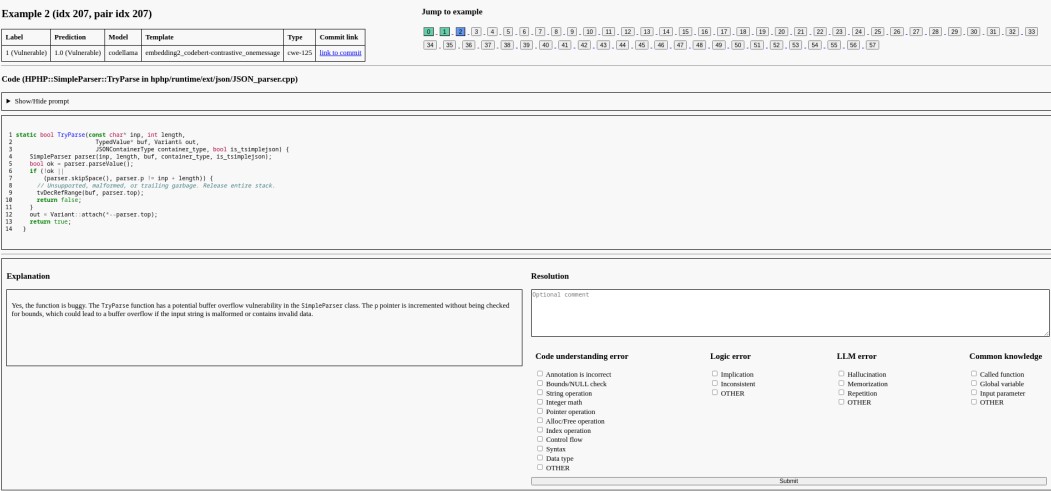

FIGURE 12. Error analysis user interface.

Figure 12 is a screenshot of the user interface (UI) used by the raters for error analysis. The interface features a display of the source code (center), the model's response and explanation (bottom left), and metadata about the vulnerability (top left). Additionally, it provides a configurable set of checkboxes to select one or more error categories, along with a section for free-form text notes (bottom right). We believe that this tool could be valuable for future large-scale manual analyses of LLM responses, which is why we have included it in our data package.

### F.3   ERROR CATEGORIES

**Filtering:** We chose the 100 shortest examples by line count in SVEN to ensure annotators could easily understand the analyzed code. We included only examples where the LLM predicted "vulnerable" to study its reasoning about vulnerabilities. We filtered for responses where the LLM provided reasoning, excluding simple answers like "Yes" or "No." To balance responses across models, we randomly excluded a small number (1–10) of responses from later-released models, StarCoder2 and DeepSeek, to reach an even total of 300. This decision also considered the cost of our rigorous manual annotation process.

**Error categories:** Table 8 shows the definitions for the error categories which we developed in our manual analysis.

| Reasoning Step | Category | Description |
|---|---|---|
| (1,2) Localizing and understanding statements related to vulnerability | Misunderstood Bounds/NULL check | Does the model state a false proposition about a bounds check or null check, such as "if (ptr) *ptr" or "if (i ¡ len) buf[i]"? |
| | Misunderstood string operation | Does the model state a false proposition about allocation, copy, reading, or writing of strings? |
| | Misunderstood arithmetic operation | Does the model state a false proposition about an arithmetic operation, such as +, -, /, *? |
| | Misunderstood pointer operation | Does the model state a false proposition about a statement involving a pointer dereference operation? |
| | Misunderstood alloc/free operation | Does the model state a false proposition about a memory allocation such as malloc or new? |
| | Misunderstood index operation | Does the model state a false proposition about a statement involving an array index operation? |
| | Misunderstood execution order | Does the model state a false proposition about a conditional, such as if or switch, or the order of execution between two statements? |
| | Improper assumption | Does the model make an unreasonable assumption about a function, variable, or parameter in the example? |
| | Misunderstood syntax | Does the model misinterpret the syntax of visible code, e.g. interpreting the declaration `int *x = NULL;` as a dereference of x? |
| (3) Logical reasoning | Faulty implication | Does the model make a logical implication where the conclusion does not follow from the premise(s)? |
| | Inconsistent | Does the model make any statements within its response which are contradictory? |
| Cross-cutting errors | Hallucination | Does the model reason about code that isn't there? |
| | Memorization | Does the model reason about code that is potentially memorized from the training data, such as talking about the calling context? |
| | Repetition | Does the model output repeated sentences which don't make sense in sequence? |

TABLE 8. Definitions of Model Reasoning Errors

