# OpenReview forum: "To Err is Machine: Vulnerability Detection Challenges LLM Reasoning"
_ICLR.cc/2025/Conference — Submitted to ICLR 2025_

### Official Review · Reviewer_qxub · 2024-10-23

**Soundness:** 2
**Presentation:** 3
**Contribution:** 2
**Rating:** 6
**Confidence:** 3

**Summary:**

In this paper, the authors performed an empirical study of using LLMs for software vulnerability detection. The authors formulated vulnerability detection in 3 steps: locating potentially vulnerable statements, understanding vulnerability-related features, and predict the final vulnerability label with multi-step reasoning. The authors conducted experiments on the SVEN vulnerability dataset with LLMs under basic and COT prompts. From the results, the authors find that existing LLMs cannot effectively detect vulnerabilities, achieving results similar to random guesses. Moreover, scaling up model parameters, fine-tuning with domain-specific data, and COT prompting do not significantly improve model performances.

**Strengths:**

+: The paper points out that vulnerability detection is a challenging task for LLMs and has not been successfully addressed by existing models. This introduces new research opportunities in both software engineering and LLM.

+: The authors defined a three-step reasoning framework for better analysis in LLMs for vulnerability detection.

**Weaknesses:**

-: In section 2, the authors adopted the BigVul and D2A datasets for building prompts. However, due to a previous study [1], these datasets have high noise rates. The authors should ensure the correctness of these generated prompts.

-: The experiments should contain some of the newest LLMs, e.g., GPT-4o-mini and Llama 3.1.

-: The authors pointed out the difficulties of vulnerability detection using LLMs. Perhaps it is also better to discuss possible solutions to these difficulties.


References:

[1]: Croft, R., Babar, M. A., & Kholoosi, M. M. (2023, May). Data quality for software vulnerability datasets. In 2023 IEEE/ACM 45th International Conference on Software Engineering (ICSE) (pp. 121-133). IEEE.

**Questions:**

- The authors mentioned the newest PrimeVul dataset, and used it for fine-tuning. So why do the authors used the SVEN dataset for evaluation instead?

- In section 3, how are the 300 manually inspected samples selected?

- In section 3.3, the authors used the COT-annotation prompts (with static analysis) to provide additional knowledge. How is the COT-annotation prompt different from other prompts in section 2, especially the COT-StaticAnalysis prompt?

---

> ### Author Response · Authors · 2024-11-21
>
> Thank you for your thoughtful review! Here we answer your questions.
>
> > Question 1: SVEN vs. PrimeVul
>
> SVEN is a manually curated dataset with rich metadata and highly accurate labels, making it the most reliable and well-understood vulnerability dataset that we know of. In contrast, PrimeVul is a large, automatically curated dataset designed for fine-tuning but lacks SVEN's rich metadata and label accuracy. As demonstrated in Table 1 from the PrimeVul authors (shown below), SVEN has more accurate labels than PrimeVul. Therefore, we believe our in-depth study of LLM responses is best conducted using SVEN and would not significantly benefit from incorporating the additional data size of PrimeVul. We will clarify this rationale for SVEN in the paper.
>
> | Benchmark            | Correct (%) |
> |-----------------------|-------------|
> | SVEN                 | 94.0        |
> | PRIMEVul-OneFunc     | 86.0        |
> | PRIMEVul-NVDCheck    | 92.0        |
>
> > Question 2: Selecting responses for manual analysis
>
> We selected the 300 LLM responses for manual observation by a filtering approach.
> * We chose the 100 shortest examples by line count in SVEN to ensure annotators could easily understand the analyzed code.
> * We included only examples where the LLM predicted "vulnerable" to study its reasoning about vulnerabilities.
> * We filtered for responses where the LLM provided reasoning, excluding simple answers like "Yes" or "No."
> * To balance responses across models, we randomly excluded a small number (1–10) of responses from later-released models, StarCoder2 and DeepSeek, to reach an even total of 300. This decision also considered the cost of our rigorous manual annotation process.
>
> We will rework Sec 3 to clarify these points.
>
> > Question 3: CoT-Annotation vs. CoT-StaticAnalysis and CoT-CVE
>
> Our COT-annotation prompt differs from other prompts in Section 2 in its use of annotations that enable us to use _lightweight, custom-built static analysis_ to provide targeted information about specific vulnerability semantics, such as bounds and NULL checks—areas where LLMs struggle, as shown in Section 3. This approach, while highly customized, requires manual effort to implement for each vulnerability type. As a proof of concept, we focused on Null-Pointer Dereference vulnerabilities, which are most strongly affected by the semantics LLMs struggled with (Table 3: bounds checks, strings, pointers).
>
> The key difference between our COT-annotation prompt and the COT-StaticAnalysis prompt is that the latter relies on _heavyweight, off-the-shelf commercial static analyzer (Infer)_ to supply proofs for the vulnerabilities Infer is designed to handle. However, it cannot provide customized information. Running Infer on arbitrary projects requires significant manual setup and infrastructure, as highlighted in the D2A paper. To avoid this complexity, we utilized existing Infer reports from the D2A dataset instead of re-running the analysis. We acknowledge the limitations of commercial static analyzers, which cannot be easily extended to address the specific semantics LLMs struggle with.
>
> In summary, to explore the research question, “How well can LLMs perform when they are provided the _best possible information_?”, we developed a lightweight static analyzer tailored to provide more customized information than Infer/D2A can offer.

---

> > ### Author Response · Authors · 2024-11-21
> >
> > > Weakness 1: data quality of BigVul and D2A
> >
> > Croft et al. highlighted the impact of dataset label quality on model evaluations. We agree on the importance of ensuring the correctness of the prompts; during development, we addressed this noise by filtering examples to include only those with viable information for accurate predictions. During development, we manually reviewed approximately 100 chains of thought to ensure that they were relevant to the corresponding code.
> >
> > - For both approaches, we removed duplicate examples, addressing a key concern raised by Croft et al.
> > - For CoT-CVE, we retained only those examples tied to vulnerabilities (i.e., bug reports) of the same types in SVEN, and excluded those with lengthy source code, making the resulting chain of thought more relevant for the model’s prediction.
> > - For CoT-StaticAnalysis, we selected examples with complete vulnerability proofs within the vulnerable function and removed those with incomplete reports, which improves overall quality.
> >
> > We will clarify these procedures in our supplementary material, which explains these approaches.
> >
> > > Weakness 2: newer LLMs
> >
> > We evaluated LLAMA 3.1 8b, as shown below, and found that it did not improve performance in vulnerability detection, achieving only 50.8 Balanced Accuracy at best. This is lower than the smaller and earlier models, Mistral 7b and Magicoder 7b.
> >
> > | Model    | Prompt                            |   BalancedAccuracy |
> > |----------|-----------------------------------|--------------------|
> > | LLAMA3.1 | Embedding                         |               50.8 |
> > | LLAMA3.1 | CoT-CVE                           |               50.6 |
> > | LLAMA3.1 | Random (Contrastive)              |               50.6 |
> > | LLAMA3.1 | CoT-StaticAnalysis                |               50.5 |
> > | LLAMA3.1 | CoT-StaticAnalysis (Contrastive)  |               50.3 |
> > | LLAMA3.1 | Embedding (Contrastive)           |               50.3 |
> > | LLAMA3.1 | Random                            |               50.1 |
> > | LLAMA3.1 | CoT-CVE (Contrastive)             |               49.8 |
> > | LLAMA3.1 | Basic                             |               48.0 |
> >
> > We will add these results to the next revision of the paper.
> >
> > Beyond the results of specific models, Table 1 highlights the challenges LLMs face in vulnerability detection, where performance has stagnated despite nearly 3x improvements on other reasoning tasks.
> >
> > In Sections 3.1 and 3.2, we explored standard scaling strategies—such as increasing training data size, model size, and incorporating code data—but found these approaches insufficient for improving vulnerability detection. This suggests that standard language model scaling may not be sufficient to tackle this problem.
> >
> > > Weakness 3: Possible solutions to the difficulties we found
> >
> > Given that scaling model size, training data, code data, and fine-tuning did not improve vulnerability detection performance, even after they improved performance on reasoning about code execution, natural-language, and math, this points to a fundamental deficiency for pretraining on large corpora of code to provide vulnerability detection capability. While scaling may intuitively seem promising, our experimental results suggest otherwise.
> >
> > To address reasoning deficiencies, we implemented CoT-Annotations as a targeted approach. Despite extensive experimentation — including seven iterations of adding information, refining prompts, and evaluating on a small dev set — this approach yielded only slight performance improvements in some cases, as shown in Section 3.3. While it mitigates certain critical reasoning failures, it remains insufficient for increasing overall performance.
> >
> > As discussed in Section 3.3, we speculate that future progress will require addressing the specific code structures and reasoning errors we found (e.g. blocking issues such as logical reasoning about relations of variables), along with other potential improvements like improved reasoning, agentic debate, and model calibration (see [our response to Reviewer dBVN](https://openreview.net/forum?id=Q0mp2yBvb4&noteId=TQfR83J74d)).

---

> > > ### Comment · Reviewer_qxub · 2024-11-26
> > >
> > > Thank you for addressing my questions. I will increase my score to 6.

---

### Official Review · Reviewer_K9Pv · 2024-11-02

**Soundness:** 3
**Presentation:** 4
**Contribution:** 3
**Rating:** 6
**Confidence:** 3

**Summary:**

This paper investigates the limitations of large language models (LLMs) in performing vulnerability detection tasks, highlighting the distinct reasoning and code understanding required for effective identification of vulnerabilities. By evaluating 14 state-of-the-art LLMs across various prompt strategies on the SVEN dataset, the authors demonstrate that LLMs generally underperform in this area, particularly with tasks requiring deep code semantics, such as handling pointer operations and bounds checks. The paper offers a comprehensive breakdown of the common failure patterns observed, emphasizing the challenges LLMs face in reliably detecting code vulnerabilities and identifying specific reasoning deficiencies in current models.

**Strengths:**

Major Strengths

1. Solid study on vulnerability detection limitations in LLMs: This paper presents a rigorous analysis of LLM performance in vulnerability detection, specifically on tasks requiring nuanced code reasoning. It identifies critical failure modes, such as incorrect handling of bounds checks and null-pointer dereferences. The authors highlight these challenges through detailed case studies, including examples like the pointer dereference and buffer overflow vulnerabilities, where models fail to correctly interpret safety checks or arithmetic constraints (e.g., the issues shown in Figures 1A and 1B).

2. Comprehensive evaluation on LLM-based vulnerability detection: The paper evaluates 14 models on the SVEN dataset[1], comparing baseline and advanced prompts like Chain-of-Thought from CVE (CoT-CVE) and static analysis-derived prompts (CoT-StaticAnalysis). The extensive prompt strategy testing, including contrastive pairs and static analysis paths, shows the impacts of these prompts on model performance. Despite attempts with diverse prompts, none of the strategies led to significant improvements beyond the random-guessing baseline, reinforcing the paper’s argument that current LLM capabilities are inadequate for complex vulnerability reasoning

3. Insightful analysis of failure patterns: The paper categorizes and quantifies specific recurring errors, including failures in bounds checks (50% of bounds-related errors) and misinterpretations of pointer or arithmetic operations. This error breakdown, as detailed in Table 3, clarifies the specific reasoning steps where models consistently fail, such as misunderstanding variable constraints and execution order. These insights offer concrete areas for improvement, particularly in recognizing key programming structures relevant to security​.

Minor Strengths

1. Excellent paper presentation: The paper is organized with logical sections and visual aids, including error distribution charts (Figure 3) and model performance comparisons (Figure 2), that help to convey complex results clearly. The figures, such as those highlighting prompt performance and specific error categories, provide readers with a quick understanding of each model’s strengths and weaknesses on vulnerability tasks​.

2. Methodical approach to prompt strategy comparisons: By systematically testing and comparing various prompt designs—including zero-shot, n-shot, CoT-CVE, and CoT-StaticAnalysis—the paper provides a nuanced view of how different prompt styles affect vulnerability detection performance. This comparison offers valuable insights into prompt engineering, especially as it reveals that even advanced prompts based on structured data (e.g., static analysis proofs) fail to achieve consistent improvements​.

3. Detailed manual inspection of model errors: The paper’s manual review of 300 LLM-generated responses allowed for a deeper understanding of specific errors that automated metrics might miss. This qualitative analysis shows that LLMs frequently misinterpret pointer safety and logical implications across multiple steps of reasoning. The authors provide concrete examples of where models misjudge safe versus vulnerable code (such as failing to recognize bounds checks on pointer dereferences), giving a more rounded view of LLM limitations in this domain

[1] He J, Vechev M. Large language models for code: Security hardening and adversarial testing[C]//Proceedings of the 2023 ACM SIGSAC Conference on Computer and Communications Security. 2023: 1865-1879.

**Weaknesses:**

Major Weaknesses

1. Oversimplied in-context learning: Although the authors assess several in-context learning techniques, they do not fully explore the design rationales for each prompt or the limitations that different choices present. For example, in the CoT-StaticAnalysis prompt, the inclusion of static analysis paths from the D2A dataset was intended to help models follow logical steps, but it lacks a discussion on why certain paths were prioritized over others. Adding such detail would clarify prompt-specific limitations and help in identifying where these prompts fell short for vulnerability detection.

2. Limited exploration of RAG-based systems for code-specific semantics: While the paper mentions that Retrieval-Augmented Generation (RAG) systems could benefit vulnerability detection, it does not explore how RAG might be applied to enhance the model’s understanding of complex code reasoning tasks. Incorporating RAG systems could allow models to retrieve relevant code snippets or documentation, potentially aiding in cases where semantic context (e.g., specific variable usage patterns or documentation on safe handling of pointers) is essential for accurate reasoning[1-2]. Exploring this could offer a deeper understanding of model limitations and point to meaningful directions for future work​.

Minor Weaknesses

1. Sampling may be biased: The manual inspection of 300 samples may introduce bias, as the authors do not fully detail the criteria used to select these samples or ensure diversity. Explaining how they selected these samples (e.g., random selection, specific focus on certain vulnerability types) would strengthen the credibility of their findings and ensure that the review represents the dataset as a whole.

2. Vulnerability types are limited: The study primarily examines vulnerabilities related to bounds checks, null pointers, and pointer handling, which limits the generalizability of the findings. Including additional vulnerability types, such as integer overflow (as shown in the simple example from CWE in Figure 10), would provide a broader view of LLM limitations in software security tasks.

3. Absence of advanced evaluation metrics: While the Balanced Accuracy metric is helpful in assessing basic performance, it does not capture important aspects like semantic accuracy or reasoning consistency. Introducing metrics that focus on these qualities would provide a more comprehensive evaluation of model performance on tasks requiring deep semantic understanding, such as detecting specific security flaws in code​.

[1] Du X, Zheng G, Wang K, et al. Vul-RAG: Enhancing LLM-based Vulnerability Detection via Knowledge-level RAG[J]. arXiv preprint arXiv:2406.11147, 2024.

[2] Fayyazi R, Trueba S H, Zuzak M, et al. ProveRAG: Provenance-Driven Vulnerability Analysis with Automated Retrieval-Augmented LLMs[J]. arXiv preprint arXiv:2410.17406, 2024.

**Questions:**

Overall, this work demonstrates a professional understanding of LLM research for vulnerability detection, offering meaningful insights into the limitations of current models and practical implications for their use in code security. I am ready to defend my assessment and may consider increasing my score depending on responses to the following questions.

1. Could the authors provide further justification for their chosen in-context learning design, including any tuning or design choices made for specific prompts?

2. What limitations did the authors encounter when expanding the scope of vulnerabilities? How could future work address this limitation?

3. Did the authors consider additional metrics for evaluating semantic accuracy or reasoning consistency in model responses?

---

> ### Author Response · Authors · 2024-11-26
>
> Thank you for your in-depth review of our paper! We’re glad to answer your questions.
>
> > Question 1: Justification for our chosen in-context learning design
>
> All of our prompt designs were based on the best performance on a small dev set or the entire SVEN dataset. We are glad to expand on our prompt design choices, since we had to truncate parts of our process from the paper because of the page limit.
>
> 1. We explored several prompt designs, beginning with basic prompts (including different phrasing such as “vulnerable” vs. “buggy”, adding “question”/”answer”, listing and defining all the possible CWE types, and different system prompts), arriving at the best-performing **Basic** prompt.
>
> 2. Then, we explored several options for adding in-context examples, such as appending all the examples in one chat-assistant message, using separate messages, sampling randomly vs. using contrastive pairs, and including from 1 to 10 examples. This resulted in two more prompting methods, **Random** and **Contrastive**.
>
> 3. Next, although we did not use this keyword, we explored retrieval-augmented generation (RAG) by using embeddings to select the most relevant in-context examples. We experimented with CodeBERT, OpenAI text-ada, and SentenceTransformers all-MiniLM-L6-v2 and chose CodeBERT for the **Embedding** approach since it performed the best.
>
> 4. Next, we explored using chain-of-thought prompts generated from the outputs of an off-the-shelf static analyzer ([Infer](https://fbinfer.com/)) and [CVE reports](https://cve.mitre.org/), resulting in **CoT-StaticAnalysis** and **CoT-CVE**. Infer reported a single bug-triggering path for each example, presumably the most relevant path based on their user testing, so we did not have to prioritize among multiple paths; see [our response to reviewer qxub](https://openreview.net/forum?id=Q0mp2yBvb4&noteId=ek2oczZiSc) for more details about other filtering we did to elicit the best possible performance from these prompts.
>
> 5. Finally, we performed our error analysis on the best-performing prompts from the above selection, and based on the finding that models most frequently missed Bounds/NULL checks, we implemented a custom static analyzer to generate high-information prompts to remedy this reasoning error, namely **CoT-Annotations**.
>
> We detail our prompts in Appendix A, but will add this additional explanation of our prompt design process.
>
> > Question 2: Limitations on the scope of vulnerabilities
>
> We leveraged the C/C++ examples in the SVEN dataset, which includes five types of vulnerabilities, including integer overflows:
>
> - CWE-190: Integer Overflow
> - CWE-476: Null Pointer Dereference
> - CWE-125: Out-of-bounds Read
> - CWE-787: Out-of-bounds Write
> - CWE-416: Use After Free
>
> Table 3 lists the taxonomy of errors which we observed in LLM responses, some of which are common to several types of vulnerabilities. For instance, LLMs frequently misunderstood string operations, which caused false positives of type CWE-476 (Null pointer dereference) and CWE-125 (OOB read).
>
> Although we do not claim that these types represent all or most vulnerabilities, these are all within [the most pervasive weaknesses in the past five+ years ](https://cwe.mitre.org/top25/archive/2023/2023_stubborn_weaknesses.html), making them extremely noteworthy, and are all extremely common in C/C++ programs. We focused on these types in order to keep our error taxonomy narrow and therefore actionable for these classes of bugs, but we believe that our methodology would be useful for addressing other important vulnerability types, following the procedure outlined in [our response to reviewer dBVN under “Future research”](https://openreview.net/forum?id=Q0mp2yBvb4&noteId=Q9Na5mNwU3).
>
> Future work could address this limitation by evaluating models on other types of vulnerabilities and expanding upon our error taxonomy to include code constructs related to the new types of vulnerabilities.
>
> > Question 3: Additional metrics for evaluating semantic accuracy or reasoning consistency
>
> While we agree on the importance of evaluating the models’ semantic understanding beyond performance metrics like Balanced Accuracy, we did not explicitly calculate the metrics you mentioned. However, we believe our error analysis in Section 3 provides a comparable level of detail.
>
> We evaluated reasoning consistency by noting logical contradictions and false implications present in LLM responses (Table 3, rows under “(3) Logical reasoning”). Our observations of factual errors in the LLMs’ responses (Table 3, rows under “(1,2) Localizing and understanding…“) indicate the relatively high error rates of LLMs, with up to 50% of responses containing at least partial inaccuracies; a follow-up analysis could divide the number of errors by the number of sentences in each response to roughly estimate the semantic error rate.

---

> > ### Author Response · Authors · 2024-11-26
> >
> > > Concern: sampling may be biased…
> >
> > We randomly sampled from the shorter samples in the dataset for manual analysis, including examples of all five vulnerability types in the dataset, which ensured that the manual analysis was tractable while avoiding bias. Please refer to [our response to reviewer qxub, under "Question 2"](https://openreview.net/forum?id=Q0mp2yBvb4&noteId=RjqMuO3i54), for a detailed description of our methodology.

---

### Official Review · Reviewer_dBVN · 2024-11-04

**Soundness:** 2
**Presentation:** 2
**Contribution:** 1
**Rating:** 3
**Confidence:** 3

**Summary:**

This paper provides an in-depth analysis of LLMs' limitations in vulnerability detection, identifying the complexity of the task and the need for future work to enhance LLM’s code reasoning abilities. It further demonstrates that common strategies for enhancing LLM performance—like increasing model size, expanding training data, fine-tuning, and leveraging domain knowledge —do not significantly improve their vulnerability detection capabilities.

**Strengths:**

- The paper studies an interesting problem and pinpoints a critical challenge with great potential.
- It provides a comprehensive analysis based on the recent LLM families and demonstrates that common strategies don’t enhance LLM performance.

**Weaknesses:**

- To my understanding, the paper pinpoints the challenge of vulnerability detection across LLMs but doesn’t clearly articulate how future work could leverage the findings presented here to further improve its performance. The main contributions aren’t clearly specified.
- The additional value of breaking down different vulnerability issues into three stages is somewhat unclear and lacks evaluation. And also why do LLMs perform notably worse on tasks involving NULL checks compared to other scenarios? Offering specific suggestions for overcoming these limitations would strengthen the paper.
- The paper can benefit from extending its fine-tuning experiments beyond StarCoder2, especially by fine-tuning the SVEN and PrimeVul datasets with a broader range of LLMs.

**Questions:**

1. Can you expand on the evaluation section as discussed above?
2. How does this work compare with other recent studies on vulnerability detection, such as Steenhoek et al. (2024)?

[1] Benjamin Steenhoek and Md Mahbubur Rahman and Monoshi Kumar Roy and Mirza Sanjida Alam and Earl T. Barr and Wei Le: A Comprehensive Study of the Capabilities of Large Language Models for Vulnerability Detection,CoRR, 2024


Minor comments:
Can you expand on the experimental setup in the Secion 3? It isn’t clear to me the evaluation methods for evaluating understanding and localizing errors.

---

> ### Comment · Senior_Area_Chairs · 2024-11-20
> **please treat [1] as concurrent work**
>
> For reasons that are a bit too complicated to get into here, I'd like to ask that you and the other reviewers consider the paper by Steenhoek et al. (2024) as concurrent work (even though it was posted about eight months ago), and not penalize this submission for a lack of novelty with respect to that paper.

---

> ### Comment · Reviewer_dBVN · 2024-11-20
> **makes sense to me 👍**
>
> replying to https://openreview.net/forum?id=Q0mp2yBvb4&noteId=P3TA6sc8cA.

---

> > ### Author Response · Authors · 2024-11-21
> >
> > Thank you for your insightful comments. We'd like to address your questions.
> >
> > > Question 1: Evaluation section
> >
> > We divided the reasoning process into three steps as a conceptual framework for understanding our results at a higher level. In our manual evaluation explained in Section 3, we evaluated specific reasoning errors (“Error” column in Table 3), each tied to a failure in a critical component necessary for correct reasoning; thus, our results are empirically grounded. Among the errors, bounds/NULL checks were the most common failure observed in LLM responses. While we report this result objectively, we lack data to explain why this occurs. We speculate that these checks are not explicitly emphasized during pretraining, leading to poor recognition by LLMs, since we observed that these checks have many variations with varying syntax.
> >
> > Future work can build on our findings, such as those in Table 3, by providing targeted information about the code structures where LLMs struggle. This could involve pretraining models not just for code completion but also for reasoning about code semantics, with a focus on the areas of weakness we identified in our study, such as bounds/NULL checks, string operations, and integer operations.
> >
> > In Section 3.3 (CoT-Annotations), we present an initial attempt to provide such targeted information. While this approach improved the reasoning behind vulnerability detection for bounds/NULL checks, it proved insufficient to significantly enhance overall vulnerability detection performance, because of the prevalence of other reasoning errors. We encourage future work to explore deeper integration of static analysis and supplemental information to address these broader challenges.
> >
> > > Question 2: Comparing with other recent studies
> >
> > [As noted by the AC](https://openreview.net/forum?id=Q0mp2yBvb4&noteId=P3TA6sc8cA), Steenhoek et al. is concurrent work. We compared our work to all known studies in Section 4 (Related Work) and demonstrated that it offers greater breadth and depth:
> >
> > - Breadth: We evaluated 11 LLMs, compared to prior studies focusing primarily on GPT-4/3.5, and explored differences in model size, training data size, and pretraining data composition.
> > - Depth: Our study analyzes LLM responses within a framework that identifies specific code structures and reasoning failures critical for vulnerability detection. Additionally, previous studies did not provide detailed or actionable error categories, which we address in our work and utilize in CoT-Annotations as (Section 3.3) a proof-of-concept.
> >
> > A lot of recent work leverages LLMs for the vulnerability detection and repair (VDR) task and implicitly assumes that LLMs can effectively accomplish the VDR task. Our foremost finding is to rigorously challenge this assumption by providing strong evidence that LLMs consistently fail at this task. We hope that the community leverages our findings, taken as a whole, to realize that the usual tactics for scaling LLMs, which have been effective on so many other tasks, are unlikely to be sufficient for VDR (ref. Sections 3.1 and 3.2).
> >
> > We cannot, of course, anticipate all the ways in which the community might rise to this challenge, but one direction that seems promising is enhancing model reasoning capabilities by training implicit chains of thought with RL, as seen in OpenAI o1 [1] (as this model was released just 3 weeks before our submission, we did not have time to thoroughly experiment with it). Other possible paths forward are agentic debate [2] and calibrating LLMs [3] for VDR. We will include these observations in our next revision.
> >
> > [1] OpenAI o1-preview. https://openai.com/index/introducing-openai-o1-preview/
> >
> > [2] Improving Factuality and Reasoning in Language Models through Multiagent Debate, Du et al., ICML 2024. https://arxiv.org/abs/2305.14325
> >
> > [3] Calibration and Correctness of Language Models for Code, Spiess et al., ICSE 2024. https://arxiv.org/abs/2402.02047

---

> > > ### Comment · Reviewer_dBVN · 2024-11-23
> > > **Thanks for the response!**
> > >
> > > Thank you for the response! I appreciate the additional details and clarifications. However, I still find it difficult to see how this work can inspire future research, which remains one of my major concerns. The proposed potential directions (such as bounds/NULL checks, string operations, and integer operations) don't seem compelling enough to me. I hope the authors can provide more insights about why LLMs can't effectively detect certain vulnerabilities and expand on the evaluation or experiments about these certain vulnerabilities as discussed in weakness #2.

---

> > > > ### Author Response · Authors · 2024-11-26
> > > >
> > > > Thank you for your thoughtful feedback. We appreciate the opportunity to highlight the potential for future research.
> > > > > Why can’t LLMs effectively detect certain vulnerabilities?
> > > >
> > > > We found that the most common culprit for LLMs’ reasoning failures was that the LLMs make statements which are untrue based on code that appears vulnerable, but don’t have the detailed knowledge to know they are false, or the ability to automatically verify them; indeed, several of these examples tripped up our manual annotators and required searching the language reference to ensure correctness. To explain this in-depth, we offer examples for several of the three most common reasoning errors:
> > > >
> > > > - **Bounds/NULL checks:**
> > > > Most commonly, the models failed to recognize that a pointer is checked and predicted “vulnerable”, which is considered a false positive. Because the pointer is used in operations which are vulnerable in other contexts (such as when there is no check), the model predicts the same failure, but fails to recognize that the check would make it safe. We included an example of this failure in Figure 4.
> > > >
> > > > - **Integer operations:**
> > > > The models often failed to reason about the behavior of common integer types; for example, Mixtral claimed that an integer operation “INT_MAX - 1” would underflow, but based on the knowledge that INT_MAX > 1, we reasoned that the operation would not underflow. We included another, different example of this error type in Figure 5.
> > > >
> > > > - **String operations:**
> > > > Models reasoned incorrectly about common string processing functions. For example, wizardcoder claimed that “strnlen on an empty string will overflow”, but this routine is safe when executed on an empty string; this led to a false positive. This reasoning could be arrived at correctly by knowing that in C, strnlen depends on the null terminator to finish reading, and that statically-allocated strings always have a null terminator.
> > > >
> > > > Based on observations like these, we conclude that the overarching reason for LLM reasoning failures is that they attempt to reason about vulnerability-adjacent patterns, but cannot do so precisely, summed up in our description of “localizing and understanding statements related to the vulnerability”. This highlights the value of incorporating additional information through precise static analysis, which, as demonstrated in Figure 9a, effectively reduced the number of missed bounds checks (although the overall performance did not improve for this prototype, due to the persistence of other reasoning errors).
> > > >
> > > > > Future research
> > > >
> > > > We believe our detailed identification of reasoning failures provides a clear and actionable roadmap for future research. Instead of broadly seeking to enhance LLM reasoning, we advocate for targeted improvements with precise metrics, such as reducing the % of missed bounds checks and % of misunderstood string operations, with the result of more trustworthy reasoning. This is not possible with any of the previous methodologies for error analysis, since they are not specific enough or tied to the code structures under analysis (compared in Section 4).
> > > >
> > > > To support this, we open-sourced our error analysis tool shown in Appendix E. Researchers can use it to quickly analyze a sample of LLM responses to measure whether and how much the LLMs improved on a specific reasoning error, as we’ve done in Section 3.3. By offering concrete metrics and practical tools, we aim to accelerate both advancements in LLM reasoning research and the adoption of reasoning-based models for vulnerability detection tasks.
> > > >
> > > > LLMs also less commonly made logical inference/contradiction fallacies, and we think these could be handled by implementing self-consistency [1]. While we think this is an important consideration, we focus our response on the errors which are tied to code constructs because this is the main contribution brought by our work, and the most common error we found in LLM responses.
> > > >
> > > > Wang, Xuezhi, et al. "Self-consistency improves chain of thought reasoning in language models." arXiv preprint arXiv:2203.11171 (2022).

---

### Official Review · Reviewer_wJY8 · 2024-11-21

**Soundness:** 3
**Presentation:** 3
**Contribution:** 2
**Rating:** 5
**Confidence:** 4

**Summary:**

The paper evaluates LLMs capability in detecting Vulnerabilities in Code. It evaluates LLMs capabilities from various aspects, for example studying whether the LLM size and training process has any effect on its capability in detecting vulnerable code patterns.

**Strengths:**

- The paper is targeting an important problem. With LLMs being readily available for various AI for Code scenarios, whether they are a good tool to use for vulnerability detection is an important problem.
- The paper targets a reasonable number of LLMs for evaluation.
- The paper first establishes that LLMs are not good at vulnerability detection, then it tries to understand where the LLMS fail, whether its localization or reasoning.
- The paper also analyzes whether the LLM size and training routine, as well as giving additional information from static analyzers matters in vulnerability detection.

**Weaknesses:**

- The paper concludes that LLMs are not great at vulnerability detection based on a study on SVEN, targeting C/C++ issues. While C/C++ is one of the important languages when it comes to vulnerability detection, previous research shows that even in code generation tasks LLMs perform better in more common languages such as python and javascript than C/C++. I'm not sure that for us to claim that LLMs are  in capable of detecting vulnerabilities, is it enough to look into C/C++ only. I expect that LLMs would perform better in easier to interpret languages such as JavaScript or on vulnerability patterns that are easier to catch such as "Hardcoded credentials.". Overall, a more fine-grained breakdown on the language and type of vulnerability is needed to better understand where the LLMs can be helpful and where they cannot be helpful.
- While I find the comprehensive study on different LLMs ability to find vulnerabilities insightful, none of the follow up experiments where able to point to a particular aspect that would explain why LLMs may not perform well on detecting certain vulnerabilities. It would be great to have some suggestions/insights on training routines, fine-tuning datasets, etc that can help improve LLMs performance.

**Questions:**

- Do the authors have a breakdown of their results in terms of vulnerability type?
- Have they repeated this experiment on newer LLMs such as Sonnet 3.5, o1, and gpt4o? If they have seen reasoning issues it would be interesting to see o1 and sonnet 3.5 results in particular.

---

> ### Author Response · Authors · 2024-11-26
>
> Thank you for your thoughtful review of our paper! Below, we address your questions and concerns in detail.
>
> > Question 1: Programming language and breakdown in terms of vulnerability type
>
> Thank you for this excellent suggestion. While expanding our study to additional languages such as Python and JavaScript would indeed be a natural direction for future work, we believe our findings provide significant value as they are.
>
> We studied the C/C++ bugs in SVEN, containing five bug types which are all among [the most stubborn software weaknesses](https://cwe.mitre.org/top25/archive/2023/2023_stubborn_weaknesses.html). Regardless of LLMs’ ability to do well on C/C++ vs. Python and JS, the prevalence of these vulnerabilities shows that our research addresses an important problem: we demonstrated that current LLMs cannot reason about five of the most common vulnerabilities in two ubiquitous programming languages.
>
> We analyzed model performance by vulnerability type and found no substantial differences between bug types — no bug type was consistently easier or harder. The performance of different bug types varied by 6.9 points at most and the best-performing model/prompt achieved 58.6 BalancedAccuracy. The results of the top three models are shown below. We will add this breakdown to the supplementary material.
>
> | model            | type    |   Best prompt BalancedAccuracy |
> |:-----------------|:--------|----------------------------:|
> | codellama        | cwe-125 |                       53.67 |
> | codellama        | cwe-190 |                       55.29 |
> | codellama        | cwe-416 |                       51.88 |
> | codellama        | cwe-476 |                       52.42 |
> | codellama        | cwe-787 |                       55.86 |
> | gpt4-turbo       | cwe-125 |                       54.53 |
> | gpt4-turbo       | cwe-190 |                       54.65 |
> | gpt4-turbo       | cwe-416 |                       53.17 |
> | gpt4-turbo       | cwe-476 |                       55.61 |
> | gpt4-turbo       | cwe-787 |                       53.57 |
> | starcoder2-ta    | cwe-125 |                       58.65 |
> | starcoder2-ta    | cwe-190 |                       52.03 |
> | starcoder2-ta    | cwe-416 |                       52.11 |
> | starcoder2-ta    | cwe-476 |                       51.69 |
> | starcoder2-ta    | cwe-787 |                       54.99 |
>
> > Question 2: Newer LLMs such as Sonnet 3.5, o1, and gpt4o
>
> We appreciate your interest in newer models. Unfortunately, we did not evaluate o1 or Sonnet 3.5 as these were released only shortly before our submission. However, our results provide a critical baseline: they demonstrate that scaling foundation model size and training data alone does not significantly improve vulnerability detection and provide strong motivation for enhancing models to strengthen their reasoning capabilities.
>
> We believe that reasoning-based models, like o1, are a promising research direction. Our findings, particularly those in Sections 3.1 and 3.2, indicate that increasing model size, training data, or incorporating more code into training data did not effectively improve detection performance. We believe that reasoning-based models have potential to improve upon the pain points we observed, especially failures to logically reason (shown in Table 3). [This aligns with findings from o1’s developers in other domains, such as advanced mathematics and gameplay reasoning.](https://www.youtube.com/watch?v=eaAonE58sLU) We welcome exploration of reasoning-based models in future research, but consider it out of scope for this which evaluates state-of-the-art “foundational” LLMs.
>
> It’s worth noting, however, that the most frequent errors stemmed from failures to localize and understand code related to the vulnerability (e.g. bounds/NULL checks were missed in 50% of responses), which requires domain knowledge that reasoning in itself may not provide. Based on this result, we believe that it will also be essential to provide additional context or scaffolding to the model, as we’ve done in our prototype of CoT-Annotations in Section 3.3.
>
> We will add this interesting discussion to the next paper revision; see also [the last paragraph of our response to reviewer dBVN, starting with "We cannot, of course, anticipate..."](https://openreview.net/forum?id=Q0mp2yBvb4&noteId=TQfR83J74d).

---

### Author Response · Authors · 2024-11-27
**Updated paper draft to reflect reviewer comments + discussion**

We thank all the reviewers for their insightful comments and suggestions. We updated the paper to address your concerns.

- We added an explanation of our prompt design process to Appendix A, ref. [K9Pv question 1](https://openreview.net/forum?id=Q0mp2yBvb4&noteId=wMycXp4E3i).
- We added the explanation of how the error analysis tool can be leveraged to Appendix F, ref. [dBVN “Future research”](https://openreview.net/forum?id=Q0mp2yBvb4&noteId=Q9Na5mNwU3).
- We added a discussion of future work on reasoning models in page 10, ref. [dBVN question 2](https://openreview.net/forum?id=Q0mp2yBvb4&noteId=TQfR83J74d) and [wJY8 question 2](https://openreview.net/forum?id=Q0mp2yBvb4&noteId=LFDLFRc8Fn).
- We added the breakdown by bug type to Appendix E, ref. [wJY8 question 1](https://openreview.net/forum?id=Q0mp2yBvb4&noteId=LFDLFRc8Fn).
- We expanded on how we handled label noise in BigVul and D2A in Appendix A, ref. [qxub weakness 1](https://openreview.net/forum?id=Q0mp2yBvb4&noteId=ek2oczZiSc).
- We expanded our rationale for using SVEN in page 4, ref. [qxub question 1](https://openreview.net/forum?id=Q0mp2yBvb4&noteId=RjqMuO3i54).
- We added an explanation of our filtering process for manual analysis to Appendix F.3, ref. [qxub question 2](https://openreview.net/forum?id=Q0mp2yBvb4&noteId=RjqMuO3i54).
- We added LLAMA 3.1 8b results to Figure 1, ref. [qxub weakness 2](https://openreview.net/forum?id=Q0mp2yBvb4&noteId=ek2oczZiSc).

---

### Meta-Review · Area_Chair_CGBv · 2024-12-23

**Metareview:**

Summary
=======
The paper evaluates the capability of LLMs in detecting code vulnerabilities, particularly focusing on C/C++ code. The key finding is that current LLMs perform poorly at vulnerability detection, achieving results similar to random guessing. Common improvement strategies like increasing model size, expanding training data, fine-tuning, and leveraging domain knowledge do not significantly enhance performance.

Strengths
=======
* One of the most comprehensive evaluations of LLMs thus far, using 14 state-of-the-art LLMs using various prompt strategies
* Good analysis of failure patterns with detailed categorization of error types
* Systematic testing of different prompt designs (zero-shot, n-shot, CoT-CVE, CoT-StaticAnalysis)

Weaknesses
==========
* Limited scope by focusing only on C/C++ vulnerabilities, potentially missing insights from other programming languages
* Insufficient exploration of RAG-based systems that could enhance code-specific semantic understanding
* Potential sampling bias in the manual inspection of 300 samples
* Lack of concrete suggestions for improving LLM performance in vulnerability detection
* Use of potentially noisy datasets (BigVul and D2A) for prompt construction
* Missing evaluation of some newest LLM models

Reasons for decision
================
The paper is claimed to be the most comprehensive evaluation to date (previous and concurrent evaluations point to the same conclusion: Current form of LLMs are limited for vulnerability detection). Its limited scope and other weaknesses are certainly the areas of improvement, especially the missing of solutions (which is, to be fair, not required for this type of evaluation paper).

**Additional Comments On Reviewer Discussion:**

Not all reviewers were very engaging the authors tried their best to address concerns raised, even thought not always successfully convinced the reviewers to change their mind. Please not that the request by the Senior Area Chair to consider the work of Steenhoek et al (2024) as concurrent need justifications because there were 8 months gaps between the two papers, and I do not know the stories behind the scene. The authors responded by claiming their top-notch comprehensiveness, which is a reasonable response.

---

### Decision · Program_Chairs · 2025-01-22

Reject